# Intensity Normalisation of GPR C-Scans

**Tess X. H. Luo [1], Wallace W. L. Lai [2,]\* and Zhanzhan Lei [1]**

[1] College of Civil and Transportation Engineering, Shenzhen University, Shenzhen 518000, China

[2] Department of Land Surveying and Geoinformatics, The Hong Kong Polytechnic University, Kowloon, Hong Kong

\* Correspondence: wllai@polyu.edu.hk

**Abstract:** The three-dimensional (3D) ground-penetrating radar (GPR) has been widely applied in subsurface surveys and imaging, and the quality of the resulting C-scan images is determined by the spatial resolution and visualisation contrast. Previous studies have standardised the suitable spatial resolution of GPR C-scans; however, their measurement normalisation remains arbitrary. Human bias is inevitable in C-scan interpretation because different visualisation algorithms lead to different interpretation results. Therefore, an objective scheme for mapping GPR signals after standard processing to the visualisation contrast should be established. Focusing on two typical scenarios, a reinforced concrete structure and an urban underground, this study illustrated that the essential parameters were greyscale thresholding and transformation mapping. By quantifying the normalisation performance with the integration of image segmentation and structural similarity index measure, a greyscale threshold was developed in which the normalised standard deviation of the unit intensity of any surveyed object was two. A transformation function named "bipolar" was also shown to balance the maintenance of real reflections at the target objects. By providing academia/industry with an object-based approach, this study contributes to solving the final unresolved issue of 3D GPR imaging (i.e., image contrast) to better eliminate the interfering noise and better mitigate human bias for any one-off/touch-based imaging and temporal change detection.

**Keywords:** ground penetrating radar; C-scan; intensity normalisation

## 1. Introduction

Ground-penetrating radar (GPR) is a widely used non-destructive testing (NDT) method for subsurface surveys and imaging because of its high resolution, non-destructive nature, and continuous contact-less measurement [1]. A-, B-, and C-scans have been, respectively, used for GPR data presentation in one-to-three dimensions. A- and B-scan images are vertical depth sections that contain the characteristics of the reflected waveform, such as signal phase, amplitude, and propagation velocity. However, a series of adjacent GPR profiles should be inspected to determine the positions and sizes of the subsurface targets. Three-dimensional (3D) C-scans have thus become increasingly popular as they assist in semantic interpretation of the subsurface in a straightforward and accessible manner in comparison with B-scans.

After decades of development, GPR 3D imaging has been widely applied to civil engineering, such as mapping underground utilities [2–4], measuring changes in the physical properties of materials [5–7], and inspecting structural conditions [8–10]. Because C-scans depicted the shape and spatial distribution properties of reflectors, they could be used to distinguish road defects from other disturbance sources with similar response textures but distinct geometry (e.g., pipelines) [11]. References [12,13] extracted the edges of anomalous regions by calculating the similarity and correlations of greyscale values of neighbouring pixels in a C-scan by binarising these values and using edge detection to extract regions where the reflected energy differed substantially from the background. Reference [14] used a grey value-based image segmentation technique to extract the local

strong reflection area from a C-scan and narrow the analysis range. These studies have illustrated that the precise inversion of the subsurface world relies on a proper C-scan imaging scheme.

Since the 3D C-scan was first utilised in the 1990s, the process of C-scan generation has been gradually standardised [15]. Traditionally, the parameters were mainly based on the experience of operators, which led to inevitable human bias in the imaging results and created difficulties in determining whether a subsurface C-scan was an accurate representation of underground reality, as the choice of parameter settings may result in completely different representations [16]. GPR representations are normally evaluated by their spatial resolution and intensity contrast, but previous studies have mainly focused on the spatial resolution, which is a combined effect of antenna wavelength, survey settings such as scan/unit and sample/scan, survey speed, geometry of reflectors, and C-scan interpolations. For instance, the relationship between C-scan spatial resolution and object parameters are quantified to develop an object-based standardised workflow for C-scan generation [17]. In addition, the spatial resolution of C-scans should not yet exceed one third of the object dimensions [18]. However, they did not consider the effect of intensity contrast in C-scans on the interpretation.

The contrast of C-scans represents reflection intensities normalised from measurements. However, the subsurface is heterogeneous and different types of reflectors may yield similar GPR reflections in the form of Rayleigh, Mie, and optical scattering [1]. In addition, GPR C-scans capture complex Mie and Rayleigh scattering while the target, manifested as optical scattering, can be overshadowed. This impedes the classification of reflectors based solely on their intensities. Over the years, significant efforts have been made to reduce noise in GPR waves, but denoising GPR data may result in the incorrect elimination of target signals since these untargeted reflectors are not considered "noise" in a radargram, but, rather, interfering features in a larger-scale survey. Although 3D GPR can produce full coverage and high-resolution measurements, C-scans remain noisy and hazy, which hinders distinguishing target objects from a complicated background. By adjusting the intensity contrast, C-scans can be optimised to emphasis on target objects and distinguish what can and cannot be imaged. However, operators may introduce subjective interpretation by manipulating the intensity contrast if no standardised references are available in the industry. A blind test was conducted by [19] on both the industry server providers and university undergraduates, and it was observed that the improper interpretation of GPR C-scan is a prevalent and serious issue. Without clear guidelines for the survey procedure, it can be difficult to determine the reliability of the GPR survey results. Therefore, in this study, our goal is to standardize a portion of the GPR survey procedure by focusing on intensity normalization, an unresolved issue of 3D GPR imaging [17].

As C-scan imaging ultimately aims to identify target objects from the invisible subsurface (i.e., semantic feature extraction), a C-scan's quality is determined by whether target objects can be distinguished from a noisy background. To optimise the subsurface image and produce more accurate results, C-scan intensity normalisation methods that highlight targets should be established. This study thus investigated the parameters that affect C-scan representation results and explored intensity normalisation methods to augment contrast and reduce noise effects. An evaluation method that integrates image segmentation and Structural Similarity Index Measure (SSIM) was proposed to assess the C-scan quality. Finally, the study quantitatively evaluated the performance of the proposed normalisation scheme, and, subsequently, proposed an object-based colourisation scheme for C-scan colourisation.

## 2. Theoretical Background

This section presents an overview of the principal theory and related work, including the physical mechanism of GPR response, colourisation theory, and information restoration methods.

### 2.1. Image Deblurring and Denoising in Optical Imaging

Several studies have considered enhancing optical image interpretability by globally preserving contrast. The blurry underwater photos could be improved by separating the low- and high-frequency components, which acceptably preserved colour edge information [20]. A method for constructing a consistent gradient field based on local luminance contrast was proposed [21]. The mapping law was optimised by a linear model based on reference-contrast mapping [22]. The gradient correlation between the input and target output greyscale maps were utilised to develop a decolourisation method [23]. A local feature network was introduced to focus on local semantic features, thereby suppressing the generation of artifacts in the process of local contrast preservation [24]. These studies have illustrated the feasibility of numerous approaches for enhancing optical images.

However, the information of GPR C-scans differs from that of optical photos because of various types of scattering and attenuation, and it remains unclear whether the aforementioned approaches are suitable for C-scan colourisation. In addition, an opaque subsurface obscures prior knowledge of what lies beneath the surface. However, it was argued that a reasonable mapping function could find a suitable greyscale with respect to human visual perception [25]. Optimised intensity normalisation, therefore, aims to maintain details of the original information by improving contrast and eliminating the effects of noise.

### 2.2. GPR Wave Scattering

GPR relies on the propagation of electromagnetic waves to survey and image subsurface areas by using a transmitter to emit a signal that penetrates the host media. Subsurface materials with different electromagnetic properties backscatter the signal, which is then recorded by a receiver. The ratio of the wavelength to the reflector radius determines the reflector's visibility. The scattering effect can be modelled in three forms: Rayleigh, Mie, and optical scattering. Rayleigh scattering occurs when the feature size is substantially smaller than the light wavelength, Mie scattering occurs when the feature size is similar to the light wavelength, and optical scattering occurs when the feature size is larger than the light wavelength [1]. Optical scattering yields optimal reflections, whereas targets are invisible in Rayleigh scattering. In practice, the dimension of urban infrastructure is far larger than the GPR operation wavelength (300–2000 MHz).

Scattering is typically, although not necessarily correctly, used to describe the deviations in the paths of Rayleigh waves owing to localised non-uniformities and presents problems for GPR imaging because it reduces the amplitudes of useful signals while increasing interfering noise. In order to bring the target event into scene, the amount of scattered energy should be minimized. Hence, the signal wavelength should be much longer than the non-uniformity dimensions $\Delta L$. According to Equation (1) [1],

$$f < \frac{30}{\Delta L \sqrt{\varepsilon}} \text{ MHz} \tag{1}$$

where $f$ refers to the central frequency, $\varepsilon$ is the relative permittivity of the host medium, then $\Delta L$ ranges from centimeters to millimeters.

Common sources of scattering include the irregular surface shape of larger buried objects, rocky soils, and gas bubbles trapped in soils, and they vary in dimensions. Hence, the speckle noises in C-scans are inevitable but can be reduced.

### 2.3. Dielectric Contrast between Objects and Host Material

An important aspect of C-scan quality is feature visibility, which determines whether the target feature can be distinguished from the background medium and is governed by the dielectric contrast (manifested as the reflection coefficient) shown in Equation (2):

$$R = \frac{\text{Reflected Amplitude}}{\text{Incident Amplitude}} = \frac{\sqrt{\varepsilon_1} - \sqrt{\varepsilon_2}}{\sqrt{\varepsilon_1} + \sqrt{\varepsilon_2}} \tag{2}$$

where R is the reflection coefficient across the two vertical interfaces and $\varepsilon$ denotes the dielectric constant/relative permittivity of the host medium.

R defines the amplitude of the reflected wave proportional to that of the incident wave. For radar waves, R can be expressed as a function of the relative permittivity on each side of the interface. If $\varepsilon_1$ and $\varepsilon_2$ are similar, most of the incident wave is transmitted through the interface, which produces a weak reflection signal. If one side of the interface possesses much smaller permittivity than the other, most of the incident wave is reflected and the reflector presents a stronger intensity than the interface without producing substantial contrast in C-scans. The $\varepsilon$ values of common media can be found in the standard released by the American Society for Testing and Materials (ASTM) [26].

However, when a GPR signal penetrates a lossy medium (e.g., clay), the amplitude decreases more rapidly with depth than in a low-loss medium (e.g., sand). Attenuation also has an important effect on the amplitude of radar waves, which directly influences the unit values in C-scans. The attenuation rate depends on the electrical conductivity of the host medium. To visualise deeply buried features, a range gain function is applied to the data to compensate for the effects of attenuation. However, each type of gain function can result in the unintentional addition of different "artificial" reflection intensities to the subsurface image, which increases the noise alongside the signal.

### 2.4. Intensity Normalisation

As C-scans only contain intensity information, greyscale images are the most suitable because they have a linear scale and therefore involve less human bias. Greyscale images have contrast ranging from black at the weakest intensity to white at the strongest and a defined greyscale space that maps the stored numeric sample values to the channel of a standard space [27]. This transformation is based on the measured properties of human vision and has no concretely defined physical rule. Importantly, the scale used in this transformation process describes the reflection contrast because the received signal intensities are transformed into greyscales.

GPR imaging typically applies a linear transformation that is suitable outside of certain circumstances that require signal exaggeration. However, when imaging a heterogeneous subsurface, linear transformation results in noisy and blurred C-scans that fail to adequately reflect the details of the subsurface environment (in terms of content and contrast; they may even cause loss of contrast and structural information) when adjacent regions have similar dielectric properties. Conversely, non-linear transformation introduces human interventions that may lead to subjective bias. Thus, it is necessary to find a non-linear colourisation method that can emphasise the target object while remaining close to reality.

## 3. Materials and Methods

This study aimed to find a suitable intensity normalisation scheme for GPR surveys by investigating the relationship between the subsurface environment and GPR C-scan representation. First, the parameters of the target objects and normalisation were identified. Subsequently, a controlled experiment based on different applications was designed to quantitatively evaluate the effects of each parameter. Finally, a generalised rule was defined by the upper and lower boundaries of each normalisation parameter.

### 3.1. Target Object Categorisation and Basic Signal Processing

To establish a quantitative relationship between the parameters of the target objects and normalisation, this study investigated previous accumulated case studies from [17]. After identifying the factors important to both feature characteristics and C-scan imaging parameters, two types of subsurface structures were designed: (a) a reinforced concrete structure; and (b) an urban underground. The C-scan intensity normalisation scheme is shown in Figure 1. For each application, two representative cases were selected to illustrate the performance of each normalisation parameter (Table 1).

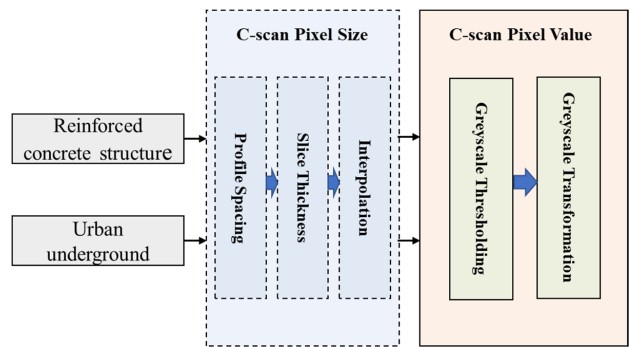

**Figure 1.** C-scan intensity normalisation scheme that continuous the 3D GPR imaging workflow.

**Table 1.** Specifications of representative cases.

| Case | Site Specification | GPR Survey Setting |
|------|--------------------|--------------------|
| | Reinforced concrete structure | |
| CW | A 1.6 m × 1.5 m reinforced concrete wall. Cover depth and diameter of buried rebar were, respectively, 0.06 m and 0.02 m. | Collected following an orthogonal grid with profile spacing of 0.1 m. Centre frequency was 2 GHz, with a 6ns time window. Frequency-domain phase-shift migration with a velocity of 0.12 m/ns was applied. |
| CS | A 3.7 × 3 m concrete slab with layers of embedded rebar. | Collected following an orthogonal grid with profile spacing of 0.1 m. Centre frequency was 1.6 GHz, with a 15 ns time window. Frequency-domain phase-shift migration with a velocity of 0.10 m/ns was applied. |
| | Urban underground | |
| UU | A section of a brick-paved road with a 0.2 m diameter drainage pipe buried underneath. The pipe was backfilled with sandy soil. Site area was 50 × 5 m. 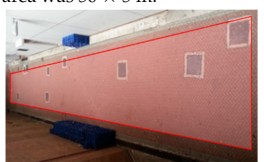 | GPR data were collected with a free loop traced by an auto-track total station. Centre frequency was 0.6 GHz, with a 30 ns time window. Range gain for consistent amplitude contrast and frequency-domain phase-shift migration with a velocity of 0.09 m/ns were applied. A bandpass filter was applied to focus on signal component of 0.37–0.82 GHz. |
| YL | A section of an asphalt paved road with three 0.2 m diameter drainage pipes located underneath. The pipe was backfilled with sandy soil. Site area was 20 m × 10 m. 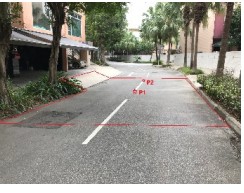 | GPR data were collected with a free loop, traced by an auto-track total station. Centre frequency was 0.6 GHz, with a 30 ns time window. Range gain for consistent amplitude contrast and frequency-domain phase-shift migration with a velocity of 0.09 m/ns were applied. A bandpass filter was applied to focus on signal component of 0.42–0.86 GHz. |

1. Reinforced concrete structure

The majority of aboveground civil infrastructures, such as buildings, roads, and tunnels, are concrete constructions. In these cases, the reflectors are defects or internal reinforcements. The ageing and degradation of concrete constructions may result in cracks,

corrosion, and water seepage. Hence, the interfaces that yield GPR reflections are dry concrete to air, metal, and wet concrete. The inner material of concrete is relatively homogeneous because the size of the aggregate in concrete or asphalt is far smaller than the GPR wavelength, which causes Rayleigh scattering that is invisible in general to the GPR.

2.   Urban underground

Many utilities are buried in the subsurface environment, including sewers, gas and water supplies, and cables. Drainage systems are concrete (plain or reinforced), water supply pipes are metal or polyvinyl chloride (PVC), and cable or gas pipelines are usually nonconducting materials. GPR signals are therefore reflected at the interface between these materials and subsurface soil at different reflection coefficient. The underground environment is complex such that unpredictable noise may hinder actual targets.

Corresponding to the cases in Table 1, the imaging parameters were adjusted and the resulting image resolution and feature reflection strength of the C-scans were observed to determine the appropriate ranges for each imaging parameter. Before constructing the C-scan, general two-dimensional (2D) basic radargram processes, other than the case-specific ones in Table 1, were conducted via the following steps: de-wow to remove the DC shift in the waveform, static correction to adjust time-zero, and background removal. The migration Envelop function was not applied because it can introduce adjacent interference. The velocity of the reflected radar waves was estimated using common offset velocity analysis, whereas the actual frequency reflected by the feature was measured using a wavelet transform [28,29]. Inspired by [30], the 2D processing was simplified to avoid the introduction of unnecessary artificial signal noise. Subsequently, C-scans were generated using GPRSLICE [31], a commercial GPR imaging software package. The B-scans were stacked into the C-scans based on a pre-designed survey grid such that each GPR signal was registered using either relative or local coordinates. The vertical and horizontal resolutions, profile spacing, slice thickness, and interpolation algorithm of the C-scans were defined based on the standardised workflow developed to ensure that the spatial resolution remained constant (10 mm for CW and CS; 30 mm for UU and YL) and the only variable was the unit value determined by greyscale colourisation in two steps: greyscale thresholding and greyscale mapping [17].

*3.2. Intensity Normalisation*

The normalisation process includes greyscale thresholding and transformation, which, respectively, record the valid reflection and intensity mapping functions. It is useful to consider a C-scan as a vector-valued image, where each unit associates with it as a vector of attribute information. The GPR survey records various noises that tend to occur at high or low frequencies. For example, large grains in host material may cause Mie reflections with high intensities in C-scans, while interference from aboveground structures results in low-intensity reflections in C-scans. The GPR signal caused by target objects and surrounding materials constitutes the main body of the C-scan histogram, thus a suitable greyscale helps to exclude this noise.

To normalise the reflection intensities, the samples of C-scans are projected onto the histogram to describe their distribution, as shown in Figure 2. The intensities are normalised on a uniform scale in the range 0–100 and a greyscale is required to match a portion of sorted intensities bounded by the low and hight cut. However, this raises two problems. First, where should the high and low cut/thresholds (i.e., the two green lines) be set and can they be set based on statistical criteria? Second, should the mapping of intensity to greyscale values be linear or non-linear? If the latter, why and how? These questions are answered in following sections.

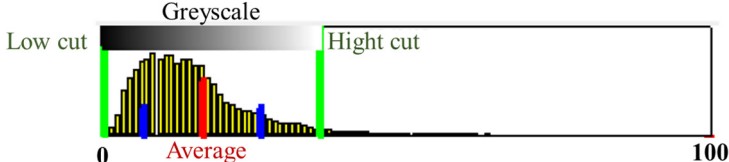

**Figure 2.** Illustration of the grayscale definition showing greyscale thresholding and transformation (captured from GPRSLICE).

1.  Greyscale thresholding

The normalised standard deviation (NSD) (Equation (3)) quantifies the range of the selected portion of the histogram where the NSD value increases with the size of the utilised portion of the histogram. The selected portion of the histogram is composed of the valid samples bounded by the high/low cut. In contrast to the standard deviation, the NSD eliminates the influence of sample values and only focuses on their distribution.

$$NSD = \frac{S}{\bar{x}} \qquad (3)$$

where $S$ is the standard deviation of the valid samples and $\bar{x}$ denotes the mean value of the valid samples.

This study evaluated C-scans with an NSD in the range 0.5–4, where 1, 2 and 3 signify respective confidence levels of 68.2%, 86.4% and 99.7% of sample values, respectively. The sensitivity test was conducted on greyscale images with an NSD step size of 0.5 such that the range of 0.5–4 was sufficiently wide for all colourisation conditions for target object categorisation.

2.  Greyscale transformation

Having selected valid sample values for the intensity range, the next step transformed the selected intensities to greyscale. Image processing techniques focus on grey-level transformations as they operate directly on units. The simplest formula for the image enhancement technique is Equation (4) [27],

$$s = Tx \qquad (4)$$

where $T$ is the transformation function, and $x$ and $s$ are the unit values before and after transformation, respectively (Figure 3).

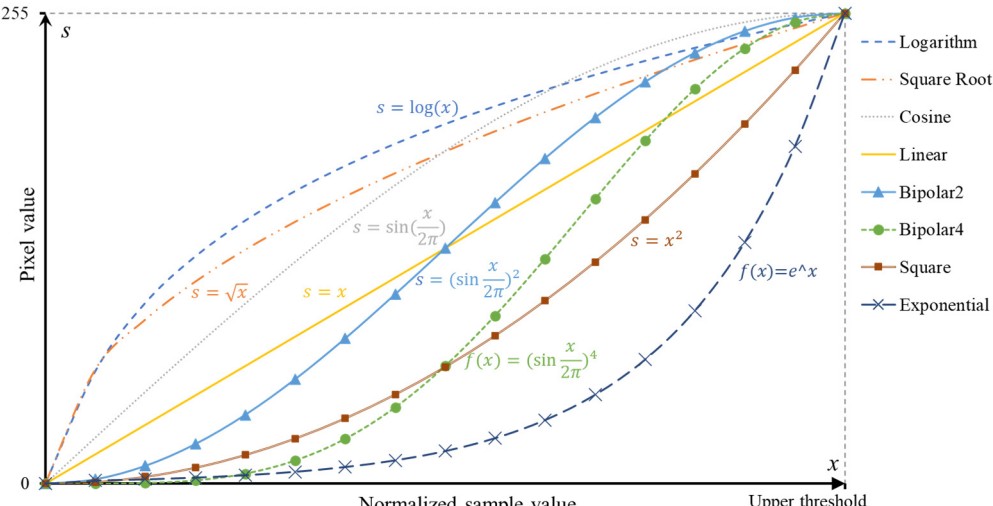

**Figure 3.** Illustration of different intensity transformation functions.

An 8-bit grey-level image involves 256 levels of grey. Figure 4 illustrates the effects of eight types of intensity mapping functions. Bipolar transformation is designed for low-

contrast cases as it allows areas of lower local contrast to gain a higher contrast. Logarithm, square root, and cosine transformations allocate more grey levels to the darker region, whereas square and exponential transformations focus on the brighter region. This study evaluated each transformation to determine their suitability for GPR applications.

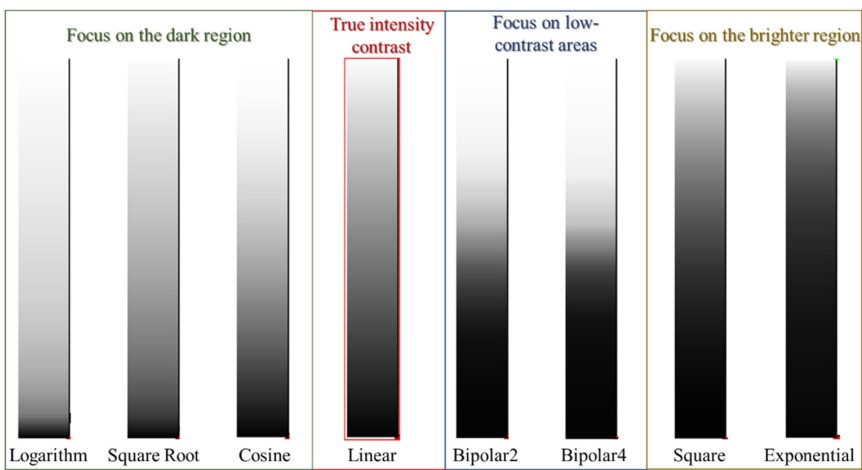

**Figure 4.** Illustration of grey level distributions of different transformation functions.

In theory, any intensity transformation can be performed in any greyscale palette model. In practice, certain transformations are better suited to specific cases. Some greyscale transformation methods are colour complements, which allows them to enhance the details embedded in the darkest or brightest portions of a grey image. Linear transformation has been widely adopted by the GPR community because its simplicity enables a true intensity contrast that reduces operator interpretation. Nonetheless, in some relatively small survey areas, buried objects appear larger than they should in a linear scale than a non-linear scale, particularly when migration is not applied to erase hyperbolic tails.

In certain cases, the dielectric permittivity of the target object is similar to that of the surrounding material, which may hide the target object if another strong reflector exists in the survey area. Subsequently, more grey levels should be allocated to the darker region to identify the weaker reflector. In contrast, if the survey area is relatively noisy and contains various strong reflectors, more grey levels should be allocated to the brighter region (i.e., optical scattering). Overall, greyscale mapping selection depends on the nature of the object.

### 3.3. Quantitative Image Evaluation

C-scan quality is difficult to evaluate. When optimising C-scan visualisation, we fine-tuned their contrast and saturations globally, based on subjective interpretation. However, high or low contrast do not necessarily refer to a "better" C-scan. Therefore, this study quantified C-scan quality by semantic interpretation performance.

In addition to C-scan quality, human perceptions also have significant influence on semantic interpretation and different professionals may provide different interpretation results from the same C-scan. To minimise human bias, this study used image segmentation method to conduct the semantic interpretation. Before choosing an appropriate image segmentation method, it is crucial to thoroughly understand the characteristics of the images. GPR C-scans are noisy and blurry, and only intensities are mapped in C-scans (low information dimension). Facing these difficulties, there are many advanced methods, and they continue developing (i.e., traditional thresholding, edge detection, region growing, deep learning-based or graph-based methods). Specifically, deep learning-based methods perform exceptional in specific cases with sufficient training. However, in the geophysical and NDT community, the number of cases used to train robust models is generally insufficient because the true labels required in the supervised approach are scarce and limited. Thus, even with a uniform algorithm or method, the correct identification of targets from

C-scans remains case-specific and it is not accurate to attribute it solely to a higher quality of the C-scan. The trained model may be more representative for one case and less so for others. The unsupervised approach is, therefore, more suitable for GPR surveys when ground truths are unavailable.

A comparative study was conducted to assess the effectiveness of various image segmentation methods suitable for blurry and noisy images, with the aim of identifying a suitable C-scan quality indicator. The selected methods included traditional unsupervised techniques such as thresholding, region growing, and active contour, as well as deep learning-based supervised methods such as U-net with level set and Mask R-CNN. These methods were evaluated on two representative cases, namely CW and UU, to extract anomalies from the surrounding background. The most stable and reliable method was deemed suitable for quantitative evaluation of C-scan quality. When anomalies were identified in C-scans, the intensity values were classified into binary scale: foreground and background. These binarised C-scans were compared with reference images created based on professional optimum semantic interpretation with the same GPR survey setting (e.g., equipment, gridding, profile spacing, slice thickness, and interpolation). The similarities between interpreted C-scans and their reference images were computed using the Structure Similarity Index Measure (SSIM) [32].

The value of the SSIM ranges from 0 to 1, where 0 signifies that the two images are independent and 1 denotes that they are identical. Therefore, a higher SSIM value indicates a better feature extraction result that is closer to the ground truth. When a better feature extraction can be produced from a C-scan, the intensity normalisation setting is considered optimal. Unlike traditional image quality measurements that estimate absolute errors, such as image differences, mean square error (MSE), or peak signal-to-noise ratio (PSNR), SSIM is saturation- and distortion-independent because it considers adjacent units to have strong interdependencies. SSIM thus emphasises the target object's structure (geometric shape) while eliminating the effects of the local minima and maxima.

## 4. Results

As shown by the reference images in Figure 5, all reflection samples were linearly transformed to grey levels to reproduce the original C-scans, which were blurry and noisy. Subsequently, active contour image segmentation was applied to extract the buried objects before manual editing was applied to refine the segmentation results based on the recorded drawings. The target objects and background materials were, respectively, labelled as white and black such that the segmented images coincided with the ground reality and served as the reference image to evaluate the performance of different intensity normalisation settings.

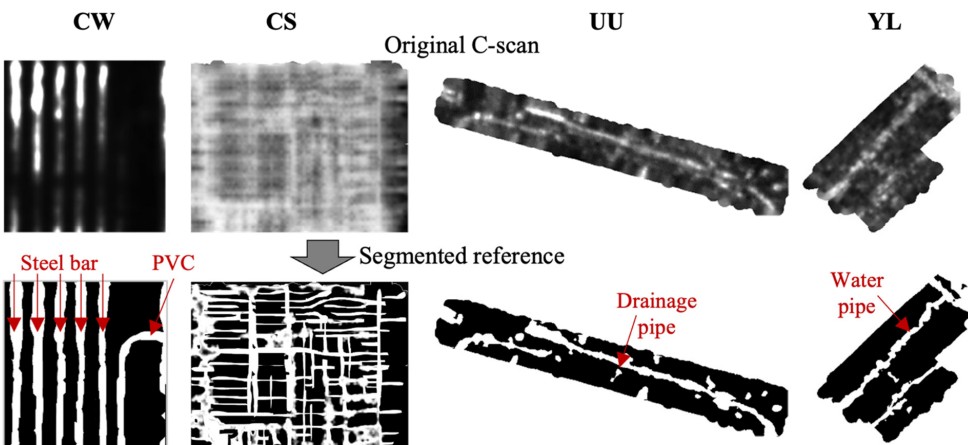

**Figure 5.** Original C-scans and segmented reference images of four representative cases named CW, CS, UU, and YL.

Five image segmentation methods were applied on fine-tuned C-scans of CW and UU. As shown in Table 2, while state-of-the-art methods may not necessarily provide more accurate results for blurry grayscale images such as GPR C-scans, this study aims to establish a general rule suitable for major civil engineering applications, including concrete structures and underground urban areas. Thus, the evaluation indicator should be stable and perform evenly in various scenarios, that is to say the accuracy and efficiency of the feature extraction method remained the same among different cases.

**Table 2.** Comparison of different image segmentation methods.

| | Unsupervised | | | Supervised | |
|---|---|---|---|---|---|
| | **Thresholding (otus)** | **Region Growing** | **Cluster-Active Contour** | **U-Net with Level-Set** | **Mask R-CNN** |
| CW | | | | | |
| UU | | | | | |

The active contour method, introduced by Kass et al. in 2D spaces, was selected to perform C-scan segmentation for GPR C-scan interpretation because it remains accurate even in images corrupted with noise [33,34]. This method is based on moving deformable contours under the forces between inner and outer energy, which helps accurately track boundaries and motions. Numerous active contour models have been developed for the purpose of segmenting images with various characteristics. Examples of such models include, but are not limited to, the Geodesic Active Contour (GAC), Chan–Vese model, Gradient Vector Flow (GVF) model, and Local Binary Fitting (LBF) model. In the present study, an optimized active contour model was developed by incorporating a clustering method to preserve its unsupervised nature. Initially, the k-Means clustering algorithm was utilized to separate image intensities into foreground and background classes, which produced a preliminary mask. The active contour model was then applied iteratively to refine the mask until it reached the convergence condition.

### 4.1. Reinforced Concrete Structure

1.  Greyscale thresholding

Table 3 presents the results of various threshold for the two reinforced concrete walls. The transformation function applied in Table 3 is linear, thus the only variable is the greyscale threshold described by NSD. For the CW case with three types of materials (cement, metal, and PVC), the five rebars were clearly visible regardless of the size of the NSD because the dielectric constant of the metal was substantially larger than that of the surrounding cement. However, the reflection of the embedded PVC pipe was relatively weak compared with that of the rebar. When the greyscale was narrowed to the highest contrast portion (NSD = 0.5), the PVC pipe was hidden in the dark (the bent linear object). Alternatively, when the NSD of the greyscale exceeded three, more intensity values were valid and assigned to darker grey levels. This reduced the contrast, which resulted in the classification of more reflections as background and the subsequent disconnection of the rebar. The SSIM evaluation coincided with the visual perception when the NSD was about 1.5. In this scenario, the automatic segmentation result had the highest SSIM value (0.69) and was thus closest to the ground truth. The second highest SSIM (0.68) value was given by NSD of 2.

**Table 3.** Visulisation results of reinforced concrete structure cases with different thresholds.

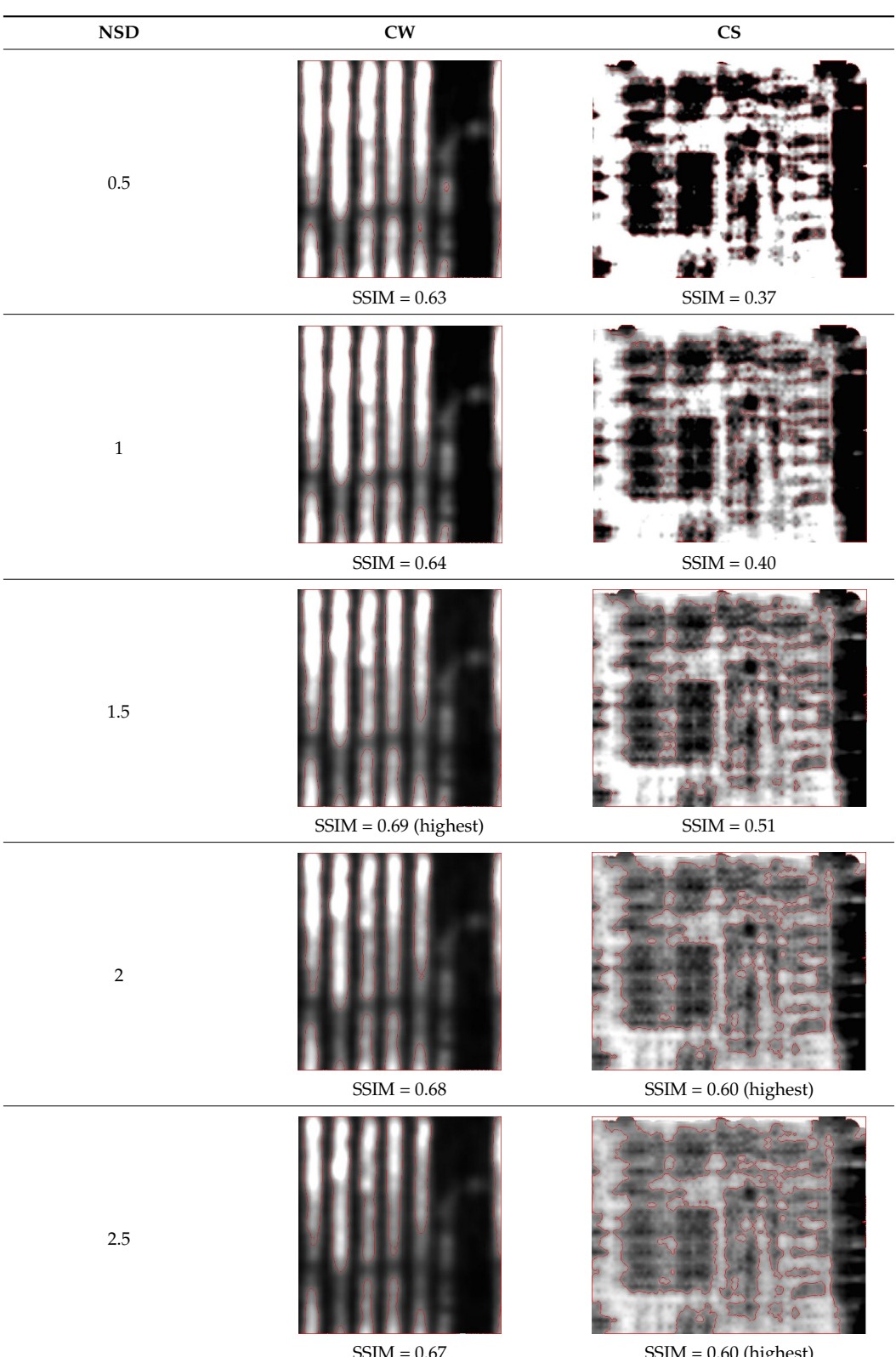

**Table 3.** *Cont.*

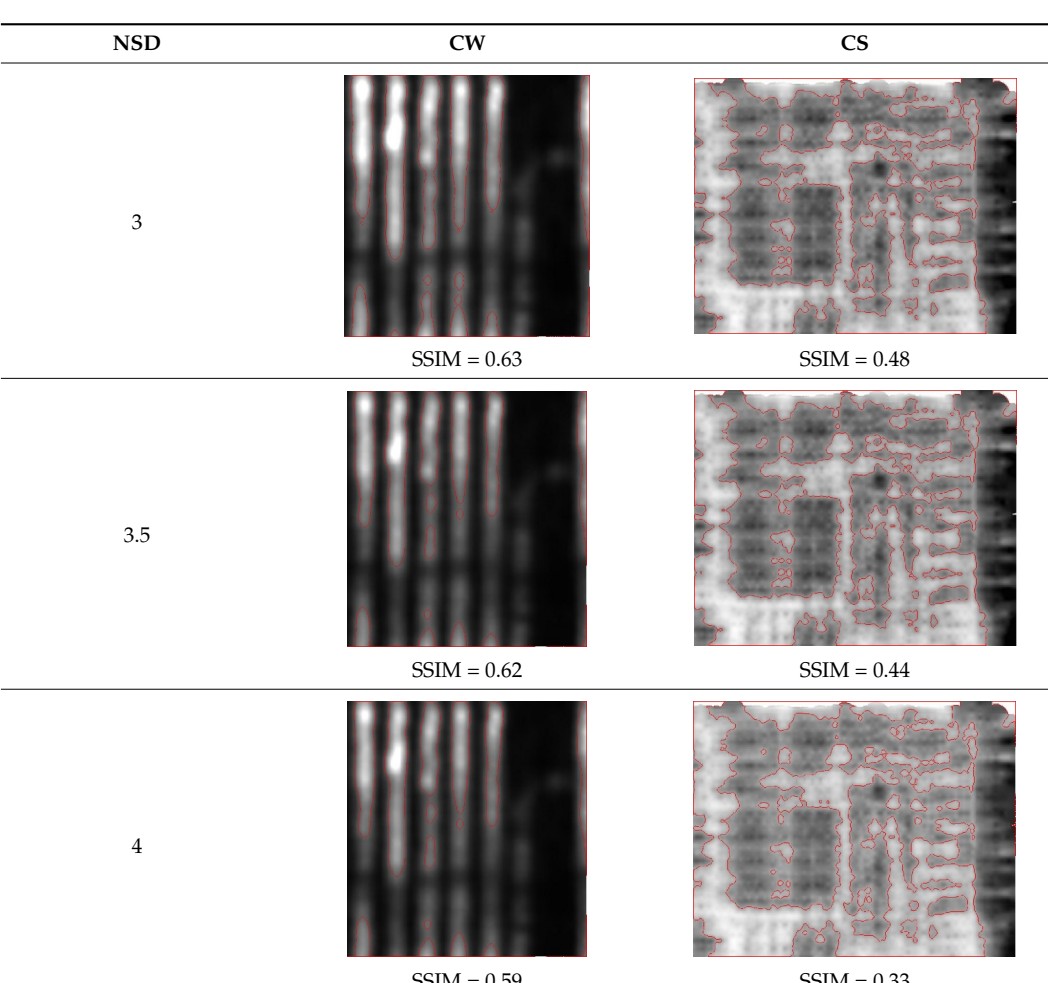

| NSD | CW | CS |
|:---:|:---:|:---:|
| 3 | SSIM = 0.63 | SSIM = 0.48 |
| 3.5 | SSIM = 0.62 | SSIM = 0.44 |
| 4 | SSIM = 0.59 | SSIM = 0.33 |

Remark: In each C-scan, black to white denote valid intensity values from weak to strong, red lines signify the automatic segmentation results, and structure similarity index measure (SSIM) values are displayed below each corresponding C-scan.

The CS case exhibited a similar phenomenon. The highest SSIM value was obtained when the NSD was 2–2.5, which indicated that 75% of the intensity values were valid. Smaller and larger NSDs both lead to a decreased SSIM, showing that the segmentation results differed from the ground truth. When the NSD decreased below 1.5, the buried rebars could not be distinguished and the two layers of mesh rebars (coloured bright white) merged into one white segment. In the same process, the single layer of mesh rebar (grey) was hidden in the dark and was recognised as the background in automatic segmentation. In contrast, when the NSD exceeded three similar grey values were allocated to the double-layer rebar and one-layer rebar. In addition, the contrast was sufficiently weak for some areas to be mis-segmented as background.

2. Greyscale transformation for civil infrastructure

Because an NSD of two outperformed in both the CW and CS cases, it was considered as the optimum greyscale threshold. When the greyscale threshold was selected, the next step defined a suitable transformation function to map the intensity to grey values. Table 4 illustrates the performance of the different transformation functions.

**Table 4.** Visualisation results of reinforced concrete structures cases with different transformations.

| Transformation | CW | CS |
|---|---|---|
| Logarithm |  NSD = 2; SSIM = 0.47 |  NSD = 2; SSIM = 0.31 |
| Square-root |  NSD = 2; SSIM = 0.50 |  NSD = 2; SSIM = 0.45 |
| Cosine |  NSD = 2; SSIM = 0.62 |  NSD = 2; SSIM = 0.52 |
| Bipolar2 |  NSD = 2; SSIM = 0.68 (highest) |  NSD = 2; SSIM = 0.57 (highest) |
| Bipolar 4 |  NSD = 2; SSIM = 0.66 |  NSD = 2; SSIM = 0.47 |

**Table 4.** *Cont.*

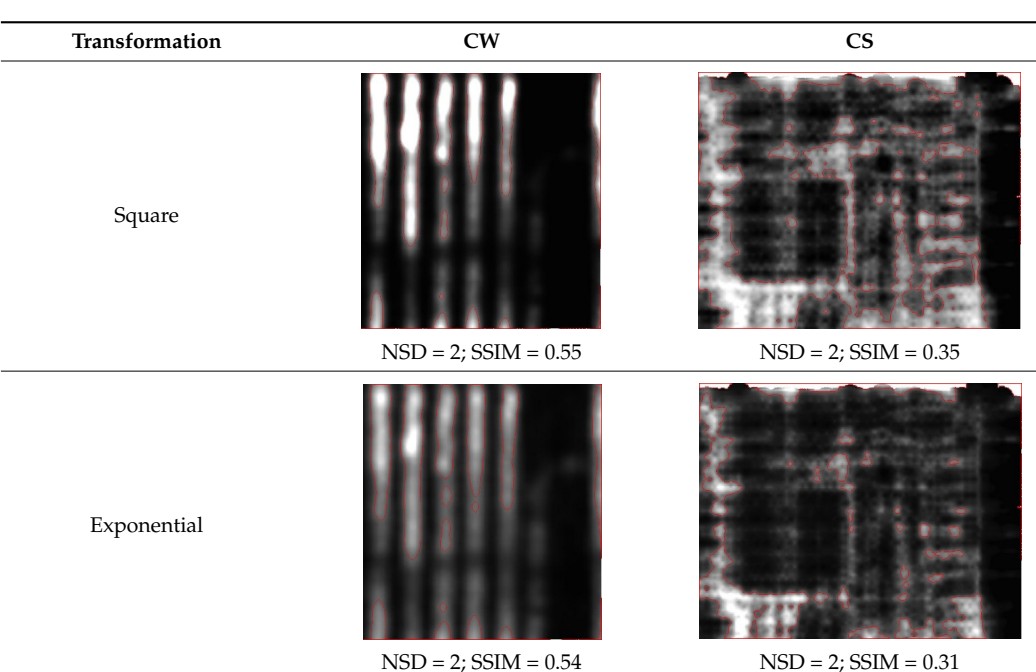

| Transformation | CW | CS |
| --- | --- | --- |
| Square | NSD = 2; SSIM = 0.55 | NSD = 2; SSIM = 0.35 |
| Exponential | NSD = 2; SSIM = 0.54 | NSD = 2; SSIM = 0.31 |

Remark: In each C-scan image, black to white denote valid intensity values from weak to strong, red lines signify the automatic segmentation results, and SSIM values are displayed below each corresponding C-scan.

For both the CW and CS cases, the optimum SSIM was given by Bipolar 2 equations. Among the seven functions, the exponential transformation provided the weakest SSIM because only strong reflections caused by the rebars were visualised as bright, while the remaining intensity values were mapped to darker grey levels. Logarithm transformation also provided unsatisfactory normalisation results as the majority of intensity values were assigned to brighter grey levels and many rebars were thus merged into one segment. For the CW case, when brighter grey levels were allocated to weaker reflections, the PVC pipe was clearer and the SSIM values, subsequently, grew. For the CS case, the SSIM values decreased when stronger intensities were transformed to darker grey levels because the mesh rebar was coloured black and incorrectly labelled as the background.

As shown in Table 4, from top to bottom, the contrast of the C-scans increased. The survey site of CW was smaller and had fewer buried reflectors, thus a smaller contrast made weaker intensities more visible. However, for more complicated cases, such as CS, stronger contrast was more suitable as the speckle noise could be more effectively eliminated. In conclusion, optimised normalisation maintained more signals for clearly highlighting target objects.

### 4.2. Urban Underground

Underground diagnosis is another essential application of GPR in urban areas, particularly as the material properties of subsurface soils are both complex and heterogeneous. Different soil types, particle sizes, buried structures, and weather can lead to diverse imaging patterns. Therefore, GPR C-scans of the urban subsurface always present speckle noises and shadow-like spots.

1.  Greyscale for urban underground

Table 5 shows the performances of different greyscales in subsurface mapping using linear transformation. The two cases (UU and YL) exhibited a similar phenomenon: along with an NSD increase from 0.5 to 4, the SSIM values first increased and then declined. Similarly to those in concrete infrastructure, the highest SSIM values were given by the NSDs of 2 and 2.5, and the second highest were provided by the NSDs of 1.5 and 2, respectively. Therefore, an average value (NSD = 2) was considered as the optimum greyscale threshold in urban underground cases. The greyscale contrast gradually decreased when

the NSD increased. When the NSD was sufficiently small, strong reflectors merged into one segment and weak reflections vanished in black. Therefore, automatic segmentation could not distinguish between the adjacent utilities. When the NSD was sufficiently large, the reflections of target object utilities were visualised at darker grey levels such that the contrast between reflectors and background was too weak to be separated by automatic segmentation. Continuous utilities were, therefore, segmented into discrete pieces.

**Table 5.** Visualisation results of urban underground cases with different greyscales.

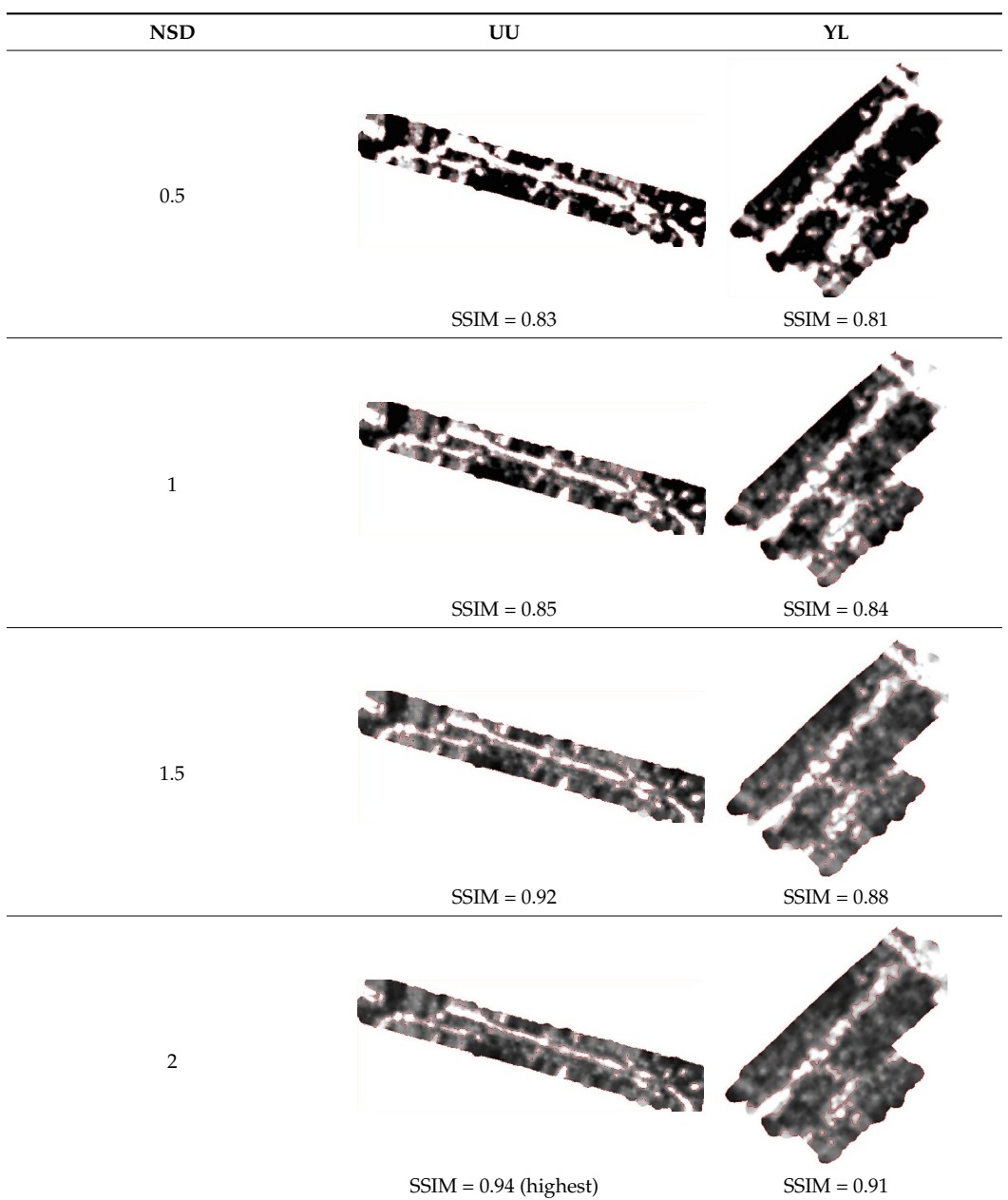

| NSD | UU | YL |
|-----|----|----|
| 0.5 | SSIM = 0.83 | SSIM = 0.81 |
| 1 | SSIM = 0.85 | SSIM = 0.84 |
| 1.5 | SSIM = 0.92 | SSIM = 0.88 |
| 2 | SSIM = 0.94 (highest) | SSIM = 0.91 |

**Table 5.** *Cont.*

| NSD | UU | YL |
| --- | --- | --- |

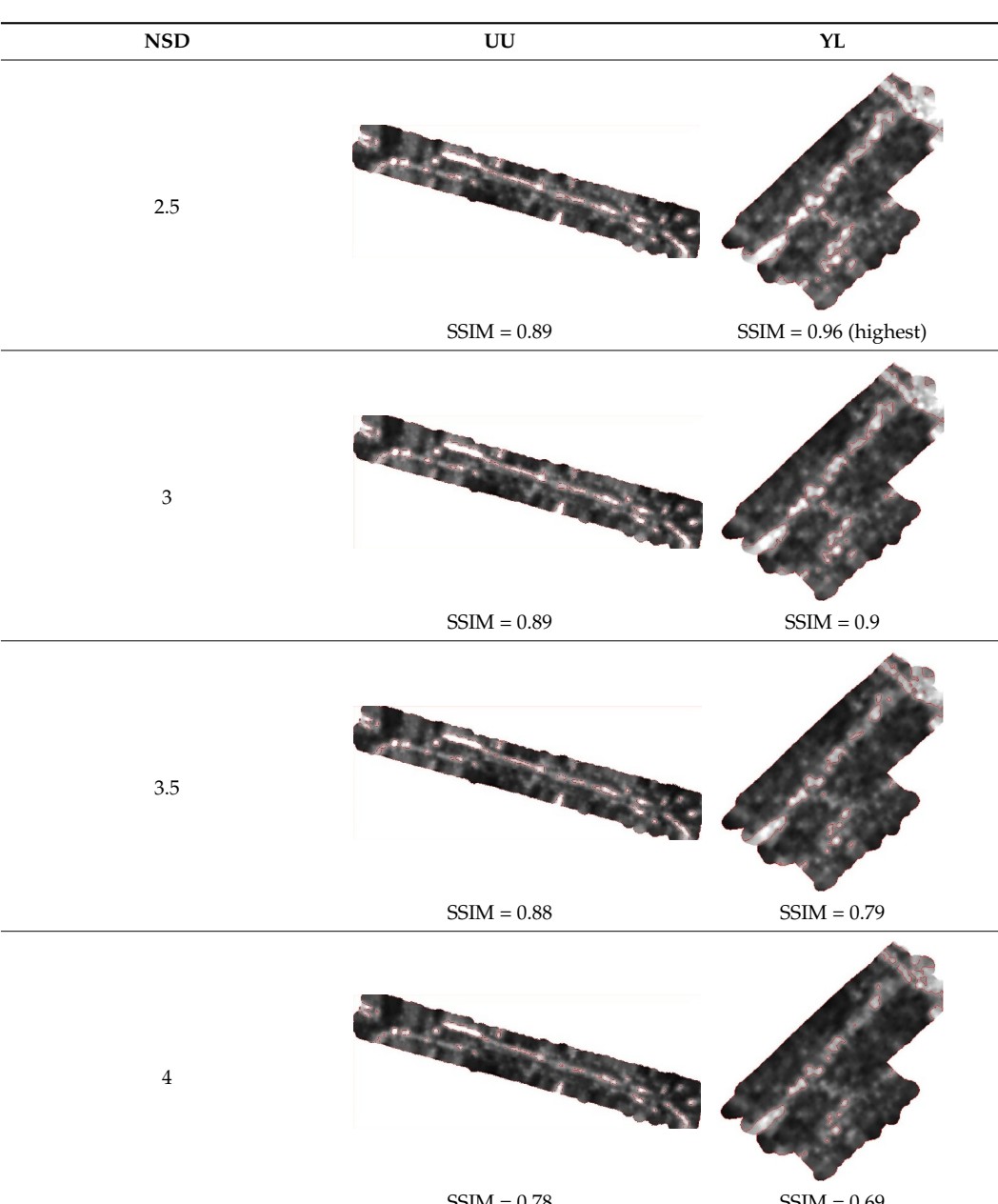

Remark: In each C-scan image, black to white denote valid intensity values from weak to strong, red lines denote the automatic segmentation results, and SSIM values are displayed below each corresponding C-scan.

Comparatively, the optimum greyscale of YL (2.5) was larger than that of UU (2). The survey grid of UU was denser than the disconnection among reflections, which facilitated interpolation. In addition, the diameters of utilities in UU were larger, causing them to be visualised as a connected linear feature. Conversely, the buried utility in YL was segmented due to the larger survey profile spacing (0.5 m). The UU and YL cases, respectively, had five manholes and one manhole surrounding the utilities, which created a simpler subsurface environment in the latter.

2. Transformation for urban underground

Table 6 lists the intensity normalisation effects of various transformations in underground applications. In both cases, the NSD of the greyscale was maintained at two and the transformation functions resulted in similar SSIM values. As in the civil infrastructure cases, the contrast of C-scans increased top down.

**Table 6.** Visualisation results of reinforced concrete structure cases with different transformations.

| Transformation | UU | YL |
| --- | --- | --- |
| Logarithm |  NSD = 2: SSIM = 0.81 |  NSD = 2; 0.85 |
| Square-root |  NSD = 2; SSIM = 0.90 |  NSD = 2; SSIM = 0.85 |
| Cosine |  NSD = 2; SSIM = 0.84 |  NSD = 2; SSIM = 0.84 |
| Bipolar2 |  NSD = 2; SSIM = 0.91 (highest) |  NSD = 2; SSIM = 0.88 (highest) |
| Bipolar 4 |  NSD = 2; SSIM = 0.85 |  NSD = 2; SSIM = 0.73 |

**Table 6.** *Cont.*

| Transformation | UU | YL |
|---|---|---|
| Square | | |
| | NSD = 2; SSIM = 0.80 | NSD = 2; SSIM = 0.71 |
| Exponential | | |
| | NSD = 2; SSIM = 0.79 | NSD = 2; SSIM = 0.69 |

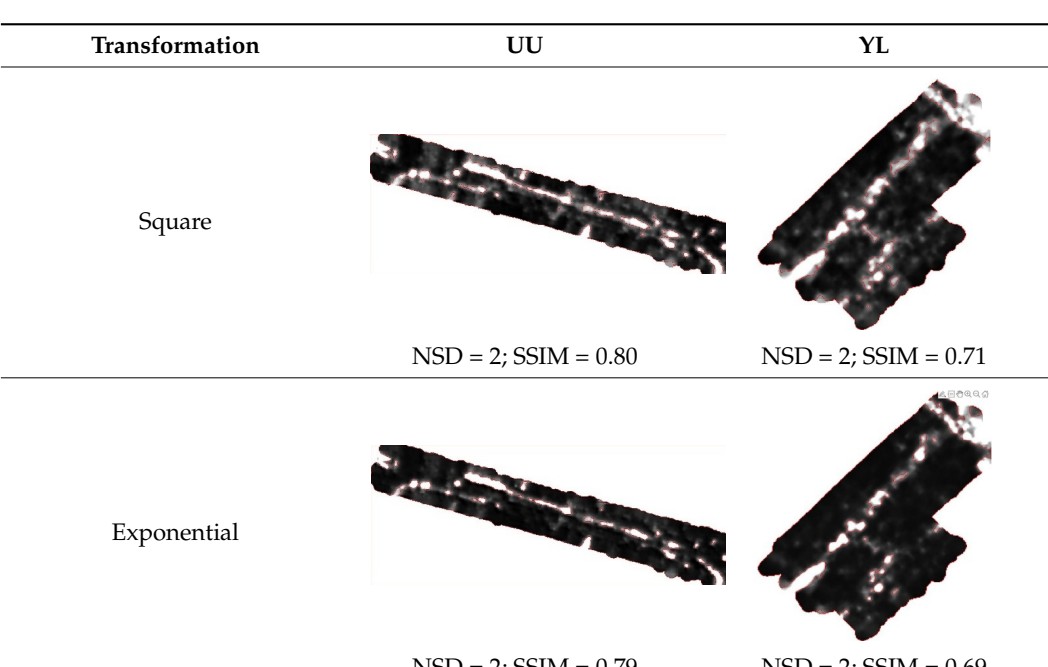

Remark: In each C-scan image, black to white denote valid intensity values from weak to strong, red lines signify the automatic segmentation results, and normalised standard deviation (NSD) and SSIM values are displayed below each corresponding C-scan.

The SSIM values of the two cases first rose and then fell, with the highest SSIM value of 0.91 (Bipolar 2) occurring in UU, which indicated that the automatic segmentation result was nearly the same as fine-tuned references. In the two underground cases, the intensities of the utilities were brightly visualised and their continuous shape was successfully depicted by automatic segmentation. The heterogeneous soils surrounding the utilities generated blurry reflections, which were visualised as grey. When the greyscale was defined with the optimum NSD value, the intensities of the subsurface infrastructure (utilities and manholes) were mapped to the bright end of the grey levels, while the remaining darker grey levels were applied to the surrounding soils. As a result, although heterogeneous soils generated different grey values with various transformation functions, they could still be classified as background. In conclusion, remarkable differences between the dielectric property of the target object and that of the surrounding material did not necessitate the exaggeration of the reflection intensity using a special intensity transformation function. Therefore, a bipolar transformation was sufficient as it involved less human intervention.

## 5. Discussion

Observation of the reinforced concrete structures and underground applications showed that the optimum setting of the intensity normalisation parameters was object-oriented, which allowed a general normalisation scheme to be derived from empirical experiments.

### 5.1. Object-Oriented Intensity Normalisation

Both for reinforced concrete structure and for urban underground cases, reducing the greyscale range enhanced C-scan contrast, which caused fewer highly intense samples to be visualised in C-scan images and thus excluded local minima and maxima. When the host material was relatively homogeneous and the type of reflectors was uniform, a stronger contrast did not result in omitting important information but, rather, contributed to emphasising the target object. Figure 6 summarises the performance of each intensity normalisation setting. For example, the optimum greyscale of the CW case had an NSD of 1.5. Conversely, increasing the greyscale threshold weakened the C-scan contrast and further resulted in the inclusion of more intense samples in the imaging. Therefore, when

the subsurface environment of the survey site was complicated and contained many types of unknown reflectors, a greyscale image with a larger NSD maintained more true reflections. Our results suggested that the NSD should exceed two, as in the YL cases. Moreover, both civil infrastructure and urban underground cases revealed that the greyscale should not be overly large (>three) or small (<one) so as to strike a balance between denoising and semantic interpretation. Overall, greyscale thresholding depends on the degree of heterogeneity in the subsurface environment.

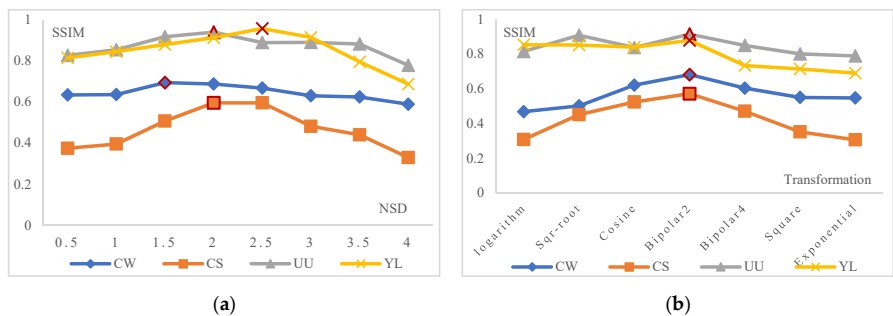

**Figure 6.** SSIM values of different (**a**) greyscale thresholding-NSDs and (**b**) greyscale transformations at the optimised SSIM = 2. Markers with red boundaries indicate optimal NSD and transformation scale. The red-bordered symbols refer to the highest values.

Selecting an appropriate transformation function could enhance the prominence of the target object and facilitate its extraction. The contrast of intensity sample values was the key factor for an optimum transformation, which relied on mapping the sample values to grey levels. If the dielectric contrast between the target object and surrounding material was significant, the noise effect was insubstantial, and the transformation function could be exaggerated. For instance, Bipolar 4 produced the second highest SSIM in CW case, with only 0.02 lower than the best SSIM produced by Bipolar 2. When the target object and the surrounding material shared similar dielectric permittivity and yielded weak reflection intensity, even transformation was beneficial as it better approximated reality. Additional grey levels could also be assigned to the darker zone to strengthen the gradient of darker grey levels and improve the clarity of the target object. As shown in Figure 6b, in UU and YL cases, the SSIM of the Square-root and Cosine are higher than that of Bipolar 4 and Square. In all cases, the Logarithm, Cosine, and Square functions yielded low SSIM values, which illustrated that using haphazard transformations in GPR coloration was inappropriate.

In addition, the SSIM of UU and YL cases are always higher than that of CW and CS. It is believed that that the variation in SSIM among cases is affected by the non-uniformities and complexities of survey sites. In the UU and YL cases, only two linear shape utilities were targeted; thus, by changing the transformation, the linear utilities were always distinguishable. In terms of the CW and CS cases, the target objects (rebars) were densely and orthogonally placed. The inconsistent orientation led to a more complex feature extraction condition. As a result, it was more difficult to realise an ideal target extraction result in the CW and CS cases.

On the other hand, the SSIM variation (standard variation, STD) of UU and YL were larger than that of CW and CS. In UU and YL, as the 600 MHz antenna was applied to arrive at the desired penetrating depth, and the minimum visible dimension was about 15.8 mm according to Equation (1), which was far smaller than the majority of subsurface non-uniformities. These subsurface scatterers varied in dielectric properties and adjusting intensity normalisation methods would change their visibility. On the contrary, in CW and CS cases, a high frequency antenna was used, and the smallest visible clutter was about 6 mm—larger than cement particle sizes. Hence, changing intensity normalisation methods would not make the scatterers visible, and the dark background of CW and CS were presented as relatively homogenous.

### 5.2. Limitations

GPR surveys have been widely applied in many fields, such as geology, archaeology, and civil engineering, but these applications likely require different intensity normalisation settings. This study investigated optimum intensity normalisation parameters, focusing on two representative scenarios: above-ground reinforced structures and underground utilities. Therefore, the scenarios discussed in this study may not be generalisable to all GPR communities. Although the two scenarios both indicate that C-scan intensity normalisation depends on the non-uniformities of the host material, establishing a universal rule requires more comprehensive cases.

In addition, the pixel values in the C-scans denoted only the reflection intensities, while the waveform information was lost. However, many reflectors have similar dielectric properties that yield similar reflection intensities, and these cannot be distinguished from the intensities of C-scans. References [29,35] verified the feasibility of using GPR reflectance frequencies in C-scan imaging. Therefore, including waveform information (e.g., velocity, phase, and frequency) in C-scans would enriching C-scan information. Moreover, a theoretical model of scattering effect contributes to eliminating undesired variation in intensity values. Notably, C-scans best serve the role of generating general overviews at different depths, thus enabling the selective interpretation of B-scans and saving efforts otherwise spent viewing every B-scan without connectivities.

### 5.3. Case Validation in Defect Detection

In addition to feature extraction, defect identification is another important purpose of conducting a GPR survey. To successfully distinguish defects from noisy backgrounds, there should be sufficient contrast, which is achieved via proper intensity normalization. The manipulation of intensity normalization may cause the survey result to become subjective. The performance of the proposed intensity normalization scheme was validated through a defect investigation case study to prove its versatility, as shown in Figure 7.

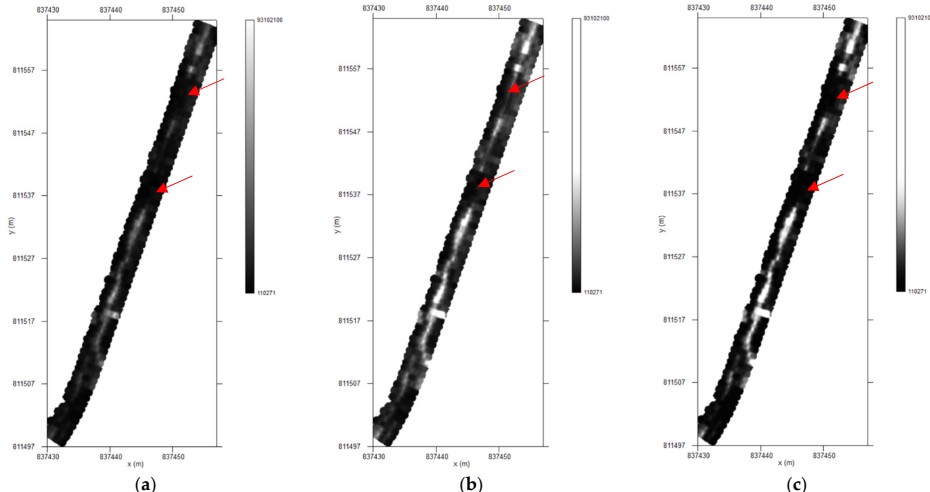

**Figure 7.** Visualised C-scans of different intensity normalisation methods: (**a**) linear transform all intensity values, (**b**) linearly transform an intensity portion of NSD = 2, and (**c**) bipolar transform a intensity portion of NSD = 2. Markers with red arrows indicate leaking points on the concrete utility.

Figure 7 presents the C-scans of a leaking concrete utility under a road surface. GPR measurements were collected using a 900 MHz shielded antenna with a spatial resolution of 0.01 m. In addition to de-wow and time zero correction, time-varying gain, a narrower frequency filter, and Kirchhoff migration with a velocity of 0.92 m/ns were applied to the GPR signals. Subsequently, the discretized intensity values were interpolated to C-scans with a spatial resolution of 0.02 m.

When all the intensity values were linearly mapped (Figure 7a), the reflection of concrete utility was weak, and the utility was visualized as segments. We could not define the wet area by disconnections. In contrast, the utility was connected to visible when only a portion of intensity values was mapped (NSD = 2, Figure 7b). However, the linear transformation resulted in global enhancement, including the wet areas. Thus, the intensity of wet soils was hindered by the subsurface scattering. By applying a bipolar transformation to a major portion of the intensity values, the reflection of utility was emphasized such that the two break points were distinguishable because of strong intensity attenuation in wet soils. Therefore, the proposed intensity normalization scheme was generally feasible in defeat extraction which are usually localised rather than continuous.

### 6. Conclusions

This study investigated the optimum visualisation for GPR C-scans in two typical scenarios: reinforced concrete structures and urban underground. Having reviewed the mechanism and principles of 3D GPR imaging, the essential parameters of intensity normalisation were identified as greyscale and transformation. An evaluation method integrates cluster-active contour image segmentation and SSIM was proposed to access the C-scan quality. Subsequently, the study quantified colourisation performance with SSIMs and determined the optimum intensity normalisation setting for each scenario. The thresholding of greyscale and transformation functions was related to the non-uniformities of the subsurface environment. This study standardised 3D GPR imaging by providing an object-based scheme for visulisation and further eliminating human bias and subjective interpretation in target object identification.

**Author Contributions:** Conceptualization, W.W.L.L.; methodology, T.X.H.L.; software, Z.L.; validation, T.X.H.L.; formal analysis, Z.L.; investigation, T.X.H.L.; resources, W.W.L.L.; data curation, Z.L.; writing—original draft preparation, T.X.H.L.; writing—review and editing, W.W.L.L.; visualization, Z.L.; supervision, W.W.L.L.; project administration, W.W.L.L.; funding acquisition, W.W.L.L. and T.X.H.L. All authors have read and agreed to the published version of the manuscript.

**Funding:** This research was founded by Research Grant Council of HKSARG and The National Natural Science Foundation of China (NSFC).

**Data Availability Statement:** Not applicable.

**Acknowledgments:** The projects "Time-lapse Imaging and Diagnosis of Urban Subsurface Hazards by Ground Penetrating Radar" with grant reference 'PolyU/15204320′ and Multi-scale and Multi-array Ground Penetrating Radar (GPR) Diagnosis of Underground Hazards 'PolyU/15216221′ funded by Research Grant Council of HKSARG, as well as the project No.42204148 founded by National Natural Science Foundation of China (NSFC) are gratefully acknowledged. The authors would also like to thank Ray Chang for his efforts in GPR data collection.

**Conflicts of Interest:** The authors declare no conflict of interest.

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
