# Peer review of "Intensity Normalisation of GPR C-Scans"

_remotesensing, doi:10.3390/rs15051309_

Round 1

Reviewer 1 Report (New Reviewer)

This paper investigates visualization techniques for GPR C-scan data and addresses a common industry challenge related to 3D GPR surveying. The topic is both interesting and of practical significance. The paper's structure and experimental design are logically constructed and comprehensive. Some minor suggestions for improvement are provided below.

1.       Introduction. It would be beneficial to discuss some recent developments in C-scan denoising. In what ways do techniques such as denoising and deblurring aid in target identification, and why is intensity normalization still necessary?

Yan, K., Zhang, Z., & Xu, X. Improved tucker decomposition algorithm for noise suppression of 3D GPR data in road detection. Near Surface Geophysics.

Feng, D., Wang, X., Wang, X., Ding, S., & Zhang, H. (2021). Deep convolutional denoising autoencoders with network structure optimization for the high-fidelity attenuation of random GPR noise. Remote Sensing13(9), 1761.

He, X., Wang, C., Zheng, R., Sun, Z., & Li, X. (2022). GPR image denoising with NSST-UNET and an improved BM3D. Digital Signal Processing123, 103402.

Hao, T., Jing, L., & He, W. (2022). An Automated GPR Signal Denoising Scheme Based on Mode Decomposition and Principal Component Analysis. IEEE Geoscience and Remote Sensing Letters.

2.       Discussion case validation. I remain unconvinced to the necessity of the validation case, as it appears to be analogous to UU/YL.

3.       Discussion. Please include future studies.

4.       There are typos and format errors (e.g. line 124, 260, equation – Equation, very-vary). Please double-check throughout the manuscript.

Author Response

Dear reviewer,

Thanks for your constructive comment, and we sincerely consider your suggestion and make responses accordingly. Please refer to the attached file. 

Reviewer 2 Report (New Reviewer)

This study investigated the optimum visualisation for GPR C-scans in scenarios related to reinforced concrete structures and urban underground. It provides a novel technique to better eliminate interfering noise and better mitigate human bias for touch-based imaging and change detection. Overall, the paper is well-structured, covers the most up-to-date literature, and presents a concrete approach validated by experimental testing. This study makes practical sense and is believed to be of great interest to readers of Remote Sensing. Some problems and suggestions are listed below.

1. The language of this paper is concise and clear. Two errors may be found: 1) in line 12 of page 1, “lead” may be changed to “leads”; 2) in line 134 of page 3, “very” should be “vary”?

        2. As the GPR survey results rely largely on the host materials, for YL case in Table 1, a brief description of the soils around the buried pipes is suggested.

Author Response

Dear reviewer,

Thanks for your constructive comment, and we sincerely consider your suggestion and make responses accordingly.

  1. Minor flaws have been corrected carefully.
  2. Soil descriptions are added to Table 1. 

Reviewer 3 Report (Previous Reviewer 3)

Dear Authors,

I was asked to review the updated (resubmitted) version of your manuscript (“Intensity Normalisation of GPR C-scan of Urban Studies”), you proposed for publication to Remote Sensing

As I already wrote, and accordingly to my previous comments, I found the manuscript is well organized and scientifically sound, the methods adequately explained, and the conclusions justified by the results. Furthermore, my last comments have been addressed, Once more, I found this research could be of interest to a wide audience, as the suggested method could be tested in other analogous fields.

I still noted some minor flaws, please refer to the attached file; particularly, check the numbering of Tables and their citation in the main text, as well as the formatting of References.

Thus, I am proposing (again) the Editor to accept the manuscript after (very) minor revisions. I hope this time your manuscript could be published.

Best Regards

The reviewer

Author Response

Dear reviewer,

Thanks for your constructive comment, and we sincerely consider your suggestion and make responses accordingly. The minor flaws have been corrected carefully.

This manuscript is a resubmission of an earlier submission. The following is a list of the peer review reports and author responses from that submission.

Round 1

Reviewer 1 Report

Dear authors,

The paper starts off well by discussing the interesting topic of scattering in GPR signals. The authors compares LUTs and applies to them an image segmentation method called active contour.

The enhancement methods in the article are basic and have been widespread since at least 1990s.

I spite of finding each day more and more unexperienced GPR operators. I consider operators should have experience and know about principles; this is not a journal for unexperienced operators.

I miss to measure the scattering as a function of the wavelength of the antenna, and to have a real theoretical model (modelled or simulated) with which to fix dielectrics, quantify the scatterings, estimate the percentage of each one and compare them, etc.

Measuring one LUT configuration regarding another by applying a standard segmentation method is poor.

The cases raised are well known to any GPR operator with some experience and do not bring new knowledge to the state of the art other than comparing different thresholds/parameters in the settings and quantifying it.

I miss for the reader a table with some theoretical calculations related to antenna wavelengths and sizes when scattering arises in these applications. I.e. from 400-600 MHz to 3,2 GHz.

Please take into account this considerations.

Author Response

Dear Reviewer

We really appreciate the efforts you put into our paper. The constructive comments and pertinent questions put forward by you provide us tremendous help to improve our manuscript. Thank you very much for your help and suggestions.

The changes of the original manuscript and replies to the reviewers’ comments are attached. Please see the attachment.

Sincerely

Reviewer 2 Report

I recommend to publish this paper in the current form.

Please cheque the spaces between "resolution" and "and" on line 56. On line 73 a verb is missing: "references" ARE "available".

Author Response

Dear reviewer,

Thanks for your comment. We have revised the manuscript accordingly. Please refer to line 56 and 73.   Sincerely Xianghuan Luo

Reviewer 3 Report

Dear Authors,

I was asked to review your work entitled “Intensity Normalisation of GPR C-scan of Urban Studies”, you proposed for publication to Remote Sensing. Actually, this is a resubmitted manuscript, and I already revised the first version (entitled “Colouring GPR C-scans”).

As in its previous form, the paper focused on the definition and testing of a specific method, in a specific field of application, and this is now clearly stated.

Accordingly to my previous review, I still found the manuscript is well organized and scientifically sound, the methods adequately explained, and the conclusions justified by the results. This new version was certainly improved, the text is better structured and more fluent, and I also noted my comments have been addressed.

I am not strictly an expert on the subject, but I found this research could be of interest to a wide audience. Beyond some minor flaws (mainly typos, or the request for some clarification: you can find my line-by-line comments in the attached file), in my opinion the manuscript can be considered for publication by Remote Sensing, after (very) minor revisions.

Again, I hope my comments could have been useful at this stage.

Best Regards

The reviewer

Author Response

Dear Reviewer,

We really appreciate the efforts you put into our paper. The constructive comments and pertinent questions put forward by you provide us tremendous help to improve our manuscript. Thank you very much for your help and suggestions.

Our response to the comments is attached, please see the attachment.

Sincerely

Xianghuan Luo

Round 2

Reviewer 1 Report

Dear authors,

The article has been significantly improved since the previous version. It now better shows certain parts that would make it possible to replicate the experiment elsewhere.

I find the article "basically" a combination of two things (LUT + Segmentation), both basic and easily parameterised, that combine to offer a range of results and see which is best.

In the most critical part, the image segmentation method, it employs none of the more current methods, which could contribute greatly to the research.

Please, see the comments.

Comment to response 1:

The authors are experienced GPR researchers who have been working extensively on GPR for decades, as well as software in GPR Slice, reflexw, Radan,  GregHD, GroundVision, etc. The author team also developed a number of software generating C-scans in LabVIEW and Python, and that is exactly why we do this manuscript because we understand the problem.

We agree that the said enhancement method is 'basic' but is far from 'widespread'. In fact, no one in commercial software nor research community specifies how the intensity scale can be objectively determined.

The reviewer is not so experienced as the authors, and has not need to use so many softwares. I basically use the hardware proprietary software (Radan and EKKO project in my case) and when I need to perform “more advanced” transformations (FDTD, denoising, change attributes of transformations, improvement in the spectra of radar signals, machine learning, etc) I must use other “open” softwares and implement it.

Current commercial software (i.e. Radan 7) permits you choose Color Tables & Color Xforms easily. The Colour Transform can be changed to enhance weak amplitude or small contrast reflectors. You can determine whether the colour scale applied to the radar wave’s amplitude is linear, logarithmic, exponential, or customized. This function can also de-emphasize certain features by pressing “just a button”.

Before in Radan 6 you could create custom LUTS (and personalised colour palettes), but they have removed this option, creating in each version the software a more "black box" model where you just click "next". Without wishing to create a debate, the latest versions of GPR software allow less and less manipulation of the data and do so in an opaquer way.

Comment to response 3:

To minimise human bias, this study used a machine learning-based image segmentation method, namely active contour, to conduct the semantic interpretation (line 312-315)

Measuring one LUT configuration regarding another by applying a standard segmentation method (active contour) is a basic experiment.

I miss at least a comparison between active contour models. It transforms the segmentation problem into the PDE framework. Snakes are computer-generated curves that move over the image to find object boundaries. The drawback of this method is that it requires user interaction. Three most commonly used PDE based methods are: Snakes, Level set, and Mumford Shah model

Current processing trends include other current “families” of segmentation methods such as Edge based segmentation (B.S.), Fuzzy theory B.S., ANN B.S., Threshold B.S., Region based B.S.

From my experience, there is not any optimal algorithm that can be applied to any kind of image. The choice of a particular techniques depends upon the application.